# User Perception of Automated Dose Dispensed Medicine in Home Care: A Scoping Review

**DOI:** 10.3390/healthcare9101381

**Published:** 2021-10-15

**Authors:** Lasse Mertz, Kristina Tornbjerg, Christian Nøhr

**Affiliations:** 1Center for Health Informatics and Technology, Maersk Mc-Kinney Moller Institute, University of Southern Denmark, 5230 Odense, Denmark; lasse.mertz@hotmail.com; 2Danish Centre for Health Informatics, Department of Planning, Aalborg University, 9100 Aalborg, Denmark; kristinat@plan.aau.dk

**Keywords:** automated dose dispensing, multi dose dispensing, primary healthcare, user experience

## Abstract

(1) Background: Automated dose dispensing (ADD) systems are today used around the world. The ADD robots are placed in patients’ homes to increase medication safety as well as medication adherence; however, little is known about how ADD robots affect the patient’s day-to-day lives, receiving the daily doses of medicine from a machine rather than from a human healthcare professional. The aim of this study is to review the available literature on users’ perceptions of having an ADD robot and collect evidence on how they perceive having less human contact after implementing this technology in their homes. (2) Methods: References were searched for in Embase and PubMed. Literature investigating ADD robots in primary healthcare was included in this study and literature in a hospital setting was excluded. After screening processes, eleven publications were included in this review. (3) Results: The literature reported high medication adherence when using ADD robots and general satisfaction in terms of user experiences with the acceptability and functionality of ADD. (4) Conclusion: The review is the first focusing on user experience and perceptions regarding ADD robots. General satisfaction was shown towards ADD robots as an intervention, but the review indicates that research is missing on healthcare professionals and patient perceptions on how ADD affects their routines, both in relation to work and daily life.

## 1. Introduction

In primary healthcare, it is a challenge to ensure medication adherence and avoid medication errors. According to a World Health Organization (WHO) report and other publications, studies have shown that, on average, 50% of people with chronic diseases do not take their prescribed medication as recommended, and, in developing countries, the number can be higher due to the paucity of health resources [1,2]. Non-adherence can be divided into two sub-groups, intentional and unintentional non-adherence. Intentionally non-adherent patients make a conscious decision to handle the medicine regime in a different way than recommended, whereas unintentionally non-adherent patients have trouble complying with the medication regime unconsciously [2,3]. Intentional non-adherence can have many different causes; it can be a result of a poor cognitive level or because of patients’ perceptions of a specific treatment. To handle intentional non-adherence, patient education can be applied to improve health literacy and provide a better understanding about a given intervention, which could increase adherence [3,4]. For unintentional non-adherence, the non-adherence can be related to demographics and clinical variables, such as socioeconomic factors, the stage of illness or the age of the patient [4].

Apart from poor medication adherence, medication errors also have an impact on adverse events within the healthcare organization. According to WHO, the number of medication errors in primary care is low but, given the number of prescriptions issued in primary care, it still can pose a serious threat to the patients who take the drugs [5]. The Danish Patient Safety Authority (DPSA) estimated that over fifty percent (55.4%) of adverse events reported in the Danish healthcare system were due to medication errors. In total, 108,111 medication errors were registered in 2018, where half of these errors occurred within home care administered by the municipalities in Denmark [6]. This number is probably larger as DPSA estimates that only 10–20% of medication errors in health organizations are being reported [6].

In many countries, there is a tendency to decentralize primary care for patients and extend the care to the patient’s own home. The Nordic countries (Finland, Sweden, Norway, Denmark and Iceland) all have a primarily tax-financed healthcare system, where the municipalities are responsible for primary care. Home care nurses who are employed by the municipalities visit patients in their homes to help them to dispense and administer their medicine. To further reduce medication errors and to save resources, municipalities are searching for technologies that can help patients to maintain high medication adherence and minimize adverse events. One of these technologies is Automated dose dispensing (ADD) robots. ADD robots are placed in the patients’ homes and deliver medication to the patient automatically without help from a caregiver, except when the medicine supply of the robot is low, in which case the caregiver will refill the robot. The robot is only used by patients who are physically and mentally capable of obtaining the full benefit of the technology. The intended goal for ADD robots is to decrease the workload of the caregiver to enable more visits to more critical patients in a day [7,8]. Studies have shown that staff within a healthcare organization tend to have different levels of burn-out or dissatisfaction with their job, and during the COVID-19 pandemic, stress factors seem have risen because of the fear of contracting the virus, but also because of the limited time available for each patient visit [8,9]. Medication robots can potentially have a positive effect on adherence and medication errors; however, to the best of our knowledge, there is no aggregate evidence of patients’ attitudes towards ADD robots—for example, patients’ perceptions about having robot technology in their home instead of daily visits from healthcare professionals. The aim of this study is to review the available literature on users’ own perceptions of having an ADD robot and to collect evidence from the literature on how they perceive having less human contact after allowing this technology into their home.

## 2. Materials and Methods

### 2.1. Literature Search

Practically identical searches were performed in December 2020 on Embase and PubMed; see Appendix A. PubMed had limited search functionality compared to Embase, but an acceptable search was achieved on both search platforms. A PICO model [10] was used to define the search strategy, where the patient was defined as a patient, civilian, citizen or user, in primary healthcare or home care; intervention was defined as an automated dose system, medication dispensing robot or multi-dose system; a comparison considered patients having medication dispensed manually or automatically, and the outcome was defined as acceptability, experience, human–robot interaction, reflection and perception of ADD robots. Sub-terms for the used keywords in the PICO model were used, to ensure a complete search. News, letters, comments, editorials and historical articles were omitted and there were no language limitations. 

### 2.2. Exclusion and Inclusion Criteria

Publications were included in the review if they investigated medicine dispensed automatically with ADD robots in multi-dose sachets for patients within primary healthcare. If publications focused on the patient’s perception of automated dose dispensation in nursing homes or their residence, or if the publication investigated medication error or medication safety with automated dose dispensation focusing on medication adherence, they were also included.

Publications investigating automated dose dispensation in hospitals and pharmacies were excluded. Studies investigating healthcare staff’s perceptions of automated dose dispensation in nursing homes or a patient’s residence were included, and papers investigating hospital staff’s perceptions were excluded. If medication adherence was not present in the study, it was excluded, and publications only available as posters were also excluded.

### 2.3. Study Selection

An iterative approach was applied for selecting the literature for this review. A web-based program, Covidence [11], was used to gather imported publications from the different literature searches, and it automatically removed duplicates. The finally included publications were imported to a reference system (Mendelay), which provides a systematic overview of all references.

### 2.4. Study Extraction

Two reviewers (LM, CN) screened the selected publications independently, based on the exclusion and inclusion criteria. A systematic overview of the included publications was performed by LM; characteristics were highlighted, in terms of study, year, country, study design, perspective and outcome measures, in a table (Table 1). The table was then reviewed by CN to ensure that the needed information was included. 

## 3. Results

The number of articles found from the search strategy was 229, of which 35 were duplicates as identified by the automatic routines in Covidence. The remaining 194 articles were then screened for both abstract and title, and 169 were excluded as irrelevant, leaving 25 articles for full-text review. In the full-text assessment, an additional 14 publications were excluded—four references appeared not to be filtered automatically as duplicates because of typing errors in reference details. A total of 11 publications were finally included for the scoping review (Figure 1). A further 10 articles were identified in a web search as gray literature but were all excluded as they had mainly a commercial purpose and background.

### 3.1. Characteristics of Included Studies

All 11 included studies were conducted in the 21st century. Table 1 shows that five study designs were represented one time: these were a scoping review, proof-of-concept study, case–control study, two-phase prospective study and systematic review. Two were qualitative studies, two were cross-sectional studies and two were quantitative studies. Of the 11 studies, ten were conducted within Europe and one was conducted in North America.

Of the 11 studies, eight explored patients with Smart Oral Multidose Dispensing Systems (SOMD), multi-dose dispensing or ADD robots, two investigated healthcare professionals’ perspectives on ADD robots, and one studied users of a smart pill bottle prototype. 

In terms of the outcome measures, six investigated patient experiences with ADD robots, eight explored medicine adherence, two investigated healthcare professionals’ opinions about ADD robots, four examined medication errors and safety when using an ADD robot, and one evaluated correct MDD patient selection.

### 3.2. Patient Experience Measures

In the six included studies that investigated patient experiences, different approaches were used in terms of data provided from the ADD robot and patient experiences from interview and questionnaires. Half of these studies considered patient experiences from the patient and from data provided from the ADD robot. The other half investigated patient experiences using interviews and questionnaires. 

Outcome specifications are shown in Table 2, where a summary of each article is provided.

## 4. Discussion

In this scoping review, a summary was generated from the included publications. A total of eleven articles were included, which indicates that research is limited within the area. Many articles were identified in the search process, but the majority were excluded from the study, because their primary focus was on general medication adherence and not with ADD robots as the intervention. The included studies investigated the user experience and patient perceptions of ADD robots and few studies investigated health professionals’ perceptions of ADD robots. A few studies investigated the acceptance and functionality of ADD robots, where patient experiences and medication adherence were highlighted [12,13,14,15,16]. One study investigated the trust towards the technology, which, when implemented, introduces new requirements, both with respect to the patient and with respect to the health professional, who are both required to adjust their daily routines around the ADD robots [17]. Two studies investigated whether ADD robots were initiated for the appropriate patients and reported that one third of the patients were possibly inappropriately initiated on ADD robots, where some patients were experiencing only a few medication problems, and that ADD robots therefore were not needed [18,19].

Overall, the included studies indicated that ADD robots as an intervention for medication management were highly accepted by both health professionals and patients. 

### 4.1. Healthcare Professionals’ Perceptions and Experience with ADD

The professionals were, in general, positive about implementing ADD robots, but a few had concerns about the trust towards the technology, regarding whether it would dispense the correct doses at the correct time. When implementing ADD robots, healthcare professionals need to be informed and trained accordingly in order for the implementation process to run smoothly without any setbacks and prevent resistance. Regarding trust, ADD robots change the workflow of the professionals, because of changes in work tasks in the environment, and trust would need to be gained through experience with ADD robots, which takes place over a longer period. For those professionals who had been working with ADD robots for a longer period, the experiences were that patient medication management was one of the most positive outcomes of using ADD robots. After the implementation phase, where work routines were changed, the health professionals perceived ADD robots as a reliable technology that provides safety for the patient [16]. Some papers reported that even though an implementation was completed successfully, some professionals still checked the patients, because their trust towards the technology had still not been established [16,17,19]. 

### 4.2. Patient Perceptions and Experience with ADD Robots

The patients were overall satisfied with their ADD robot, where most patient experience studies reported increased medication management and medication adherence [20]. Patients reported that they had better control of their own medication after receiving an ADD robot, which could be because the technology provided structure to their medicine dispensing. For some patients, this meant that they were able to actively take control over their own treatment, as the ADD system contained their specific medication and dispensed it at fixed times every day. A study from 2007 mentioned that many patients from the included patient group experienced non-compliance. However, this was only experienced when the patients were manually taking medication from multi-dose sachets at home [21]. This study was published in 2007 and the ADD technology has progressed since then, and user issues with ADD robots have gradually changed in characteristics. Some of the more recent studies mentioned that not all selected patients were suited for the technology, while others had not lost their capacity to administrate their own medication but, nevertheless, they were initiated on an ADD robot [18,19,22].

Patients were overall satisfied with the functionalities of their ADD robot. According to some of the studies, only small design flaws were of concern among the patients and one of the main issues was the visibility of printed text on the medicine sachets, where the text was hard to read. Consequently, a number of patients experienced difficulties in reading which medication they were receiving. Several patients reported that they found it difficult to open the medicine sachets, while others had to be instructed multiple times, to be able to use the ADD robot. These appeared to be start-up issues in the initial phase of the implementation. ‘Ease of use’ was mentioned in the included publications as different patient groups reported that this technology was easy to use, and they relied on the dispenser to handle their medication. 

### 4.3. Future Research

After the review of the included studies, different gaps were found in the literature, including missing information on how professionals felt about the changes in their work routines after implementing ADD robots. The studies focused on the perception towards ADD robots as an intervention and on patient outcomes. For patients, there was no information regarding experiences or perceptions on how it had affected them after incorporating an ADD robot into their daily life, and how it was perceived before versus after the implementation. This is relevant to investigate in future studies. The included publications did not investigate how ADD robots changed the environment of the users when implemented. Currently, there is no available literature within this field, which highlights the need for further research within the area. 

### 4.4. Strengths and Limitations

While other reviews have investigated ADD robots with a focus on medication adherence, the strength of this review is the focus on patient experiences and changes in the environments and settings in which the ADD robots are included. However, the limitation of this scoping review is that it is only based on literature searches in two databases and therefore may not have identified all relevant articles within the area of perceptions of ADD. 

## 5. Conclusions

This is, to the best of the authors’ knowledge, the first scoping review focusing on user experience and perceptions of ADD robots. The focus seems to be on patient medication adherence as well as patient convenience, where ADD robots were reported as being accepted in general and patients were satisfied about using ADD robots to manage their medication. The relatively low number of studies identified in this review indicates that it is a new area of research. The references found also show that the main activities until now have been concentrated in Nordic countries.

## Figures and Tables

**Figure 1 healthcare-09-01381-f001:**
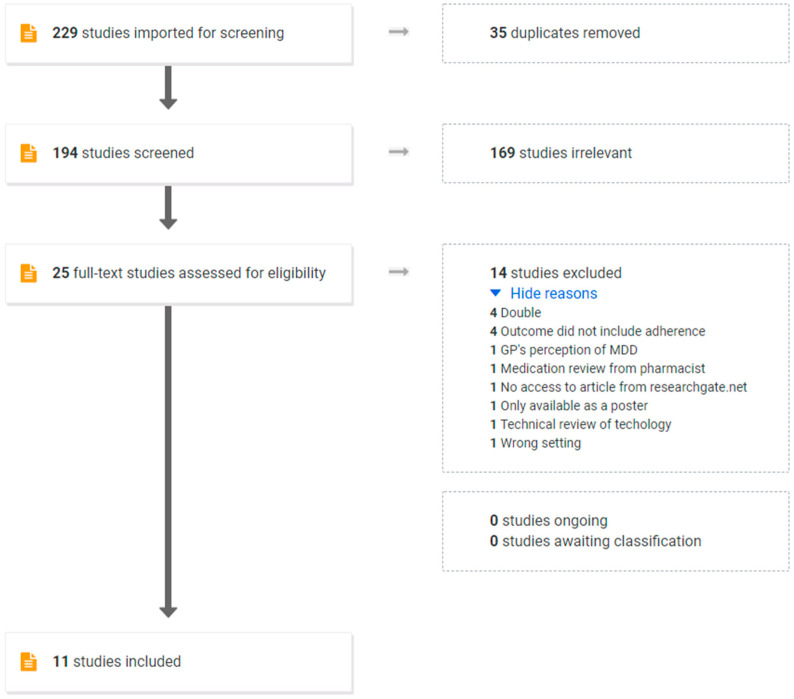
An overview of the literature retrieval process.

**Table 1 healthcare-09-01381-t001:** List of studies included in scoping review.

Study	Year	Country	Study Design	Perspective	Outcome Measures
Faisal, S. et al.	2020	Canada	Scoping review	Patients with SOMDS	Acceptability, functionality and usability of ADD robots, medicine adherence
Zijp, T. R., Touw, D. J. and van Boven, J. F. M.	2020	Netherlands	Proof-of-concept study	Users of smart pill bottle prototype	Medicine adherence, user acceptability of ADD robots
Mertens, B. J. et al.	2019	Netherlands	Cross-sectional study	Patients with MDD	Medication safety and adherence, patient’s convenience
Mertens, B. J. et al.	2018	Netherlands	Case–control study	Selection of MDD users	Correct patient selection for MDD, medication adherence
Rantanen, P. et al.	2017	Finland	Two-phase prospective study	Elderly patients’ use of ADD robots	Medication adherence, patients’ experiences
C, B. and L, R.	2016	Sweden	Quantitative study	Patients with ADD	Medication adherence, patients’ experiences
Bardage, C., Ekedahl, A. and Ring, L.	2014	Sweden	Cross sectional study	Healthcare professionals’ perspectives on ADD robots	Medication adherence, medication errors, healthcare processionals’ opinions
Sinnemäki, J. et al.	2013	Finland	Systematic review	Patients with ADD robots	Medication safety, medication use and medication adherence
Wekre, L. J., Melby, L. and Grimsmo, A.	2011	Norway	Qualitative study	Healthcare professionals’ perspectives on ADD robots	Healthcare professionals’ opinions, medication errors
Larsen, A. B. and Haugbølle, L. S.	2007	Denmark	Qualitative study	Patients with ADD robots	Medication compliance
Bredenberg, S. et al.	2003	Sweden	Quantitative study	Patients with ADD robots	Patients’ opinions

SOMDS: Smart Oral Multidose Dispensing System; MDD: Multi-dose dispensing; ADD: Automated dose dispensing robot.

**Table 2 healthcare-09-01381-t002:** Outcome specification and description of results from included studies (N = 11).

Study	Data Collection Method	Outcome Specification and Results
**Patient experiences**		
Faisal, S. et al. (2020)	Literature search	The study highlighted SOMDS as easy to use and only minor usage issues were reported. SOMDS seemed to enhance medicine adherence in various patient populations and clinicians were able to address causes of intentional non-adherence because of real-life monitoring, in spite of variety in SOMDS.
Zijp, T. R., Touw, D. J. and van Boven, J. F. M. (2020)	Questionnaire data from ADD	The study tested medication adherence (using human factor testing and product evaluation) to elucidate user acceptance and technical robustness of the prototype. Ten volunteers, who used the smart pill bottle prototype with placebo pills for 14 days, were included in the study and most of them found the system easy to use.
Mertens, B. J. et al. (2019)	Interview Questionnaire	The study explored patients’ experiences with the initiation and use of MDD systems. A total of 62 patients, aged 74–85 years, were included in the study; all of them were using MDD. The patients were satisfied with MDD as a tool to support and administrate their medication. Almost 50% reported decreased medication errors as a result of using MDD, while 14% reported having higher medication adherence, because they were notified when the medicine was ready to be taken.
Bredenberg, S. et al. (2017)	Observation Questionnaire	The study examined twenty patients’ perceptions of using MDDs. They were positive about the concept, but some found the dispenser too large, while others reported that some of the MDD features were too small. The results showed that 19 of the 20 participants were positive about the concept of dose administration in general and emphasized needs for the dose dispensing concept.
Rantanen, P. et al. (2017)	Observation Interview Questionnaire Data from ADD	The study investigated whether an automated medication dispenser had any malfunctions, when used in patients’ homes, by researching 17 patients using an automated medication dispenser for 457 days (phase one). In phase two, 27 patients used the same medication dispenser for 727 days, and the use of the robot was studied under a less controlled setting in the patients’ homes. More than 20% of the patients in both phases reported that they had difficulties remembering their medication. The study found that both patients and nurses would recommend (or probably recommend) the robot for further use. One patient responded that he would not, because of the size of the robot.
C, B. and L, R. (2016)	Questionnaire	The study used a questionnaire to gather information from a total of 1465 respondents. Overall, 58% had used an ADD robot for 2 years or longer; 53% were women, 47% were men and 64% were 65 years or older. In total, 93% of the respondents said that the ADD robot helped them to take the correct dosage, while 90% felt secure about using the ADD robot and 78% said that the sachets were easy to open. Moreover, 87% said that they disagreed with being displeased with receiving medication in sachets. In general, the patients were satisfied and felt secure using the ADD robot, and the majority would recommend ADD robots to others. One of the issues with ADD was that the patients felt that they needed better information about their treatment and about treatment-related changes. Adherence and safety issues were suggested to be investigated.
**Healthcare professionals’ experiences**		
Bardage, C., Ekedahl, A. and Ring, L. (2014)	Questionnaire	The study researched Swedish healthcare professionals’ perceived experiences with ADD robots and their effects on patients’ adherence and safety. Responses from a total of 223 physicians, 215 nurses and 915 assistant nurses were aquired and most nurses (90%) and assistant nurses (74%) stated that ADD robots increased patient safety, and 59% of physicians responded likewise. The majority of the respondents stated that ADD robots were for patients who were not capable of managing their medicine themselves and that ADD robots were offered for the convenience of the staff. Overall, 60% of the physicians, 52% of the nurses and 31% of the assistant nurses stated that patients received ADD robots to improve medication adherence and one-third of the nurses and assistant nurses replied that an ADD robot was suitable for those patients who were on stable medication, where the prescription did not change often. Nurses and nursing assistants stated that more patients should be offered an ADD robot, but because of the physicians’ negativity towards the system (for economic reasons), not all patients are offered the technology.
Wekre, L. J., Melby, L. and Grimsmo, A (2011)	Interview	The aim of the study was to investigate early experiences, including trust, between different groups of healthcare professionals participating in the implementation of ADD robots. The majority of the participants expressed positive attitudes towards the ADD robot in relation to trust, while a few felt the need to check the drug packages from the pharmacy themselves, not relying on the robot to secure the right dose. The study found that healthcare professionals’ attitudes towards a new system can affect the implementation process and outcome.
**Medication safety and use**		
Sinnemäki, J. et al. (2013)	Literature search	The study reviewed the evidence for the influence of ADD robots on appropriateness of medication use, safety and costs within primary healthcare and included 7 studies. The reviewed controlled studies suggested that patients using ADD robots were experiencing more inappropriate drug use than patients receiving standard dispensed medicine. The technology could have disadvantages in terms of continuing unchanged drug treatment, and to prevent this, more frequent medication reviews would be required from the general practitioner. Two of the four controlled studies indicated that patients using ADD robots and had more complicated drug regimens and high-risk medications were a segment who could benefit from ADD robots, as this technology could prevent higher-risk drug-related problems, medication errors or inappropriate drug use.
Mertens, B. J. et al. (2018)	Interview	The study researched and compared patients who were about to start on MDD and those who used manually dispensed medication, in order to identify whether it was the appropriate patients who received an MDD system. Overall, 418 patients were included in the study, where 188 were MDD users and 230 were non-MDD users. The study showed that the majority of the MDD systems were initiated for patients who had a decreased medication management capacity. The users of MDD were aged 76–85 years; some were cognitively impaired and frail. Of the users who were about to start on MDD, 30% were in the category of being inappropriate to start on MDD and this group stated that they had not lost the capacity to manage their own medication.
Larsen, A. B. and Haugbølle, L. S. (2007)	Interview	The study explored how the ADD robots affected users’ handling and consumption of medication, related to compliance behavior. It further investigated how healthcare professionals’ assumptions of user benefits were alligned with users’ experiences with ADD robots. Nine patients were interviewed, and 7 of these showed noncompliance. The study found that users were worried about drug dependency, about the influence of previous experience from another sector (where the same medication was used in another way) and lack of motivation to take the medication as prescribed.

## Data Availability

All data are presented in the article.

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
