# Peer review of "User Perception of Automated Dose Dispensed Medicine in Home Care: A Scoping Review"

_healthcare, 2021, doi:10.3390/healthcare9101381_

Round 1

Reviewer 1 Report

Thank you for the opportunity to review this paper, reporting a scoping review of users’ perception of automated dose dispensed medicine.

There are two main issues with the paper.

The first is that the authors do not adequately describe the setting within which automated dosing takes place. For many readers, automated dose dispensing is not available in patients’ homes, nor do healthcare workers go to people’s homes to provide medicines and possibly ensure that these are taken. This needs to be explained to avoid readers being confused, as I was. On first reading the Introduction, I wrote the following comment: “Introduction, para 3: the authors need to explain in more detail what they mean by automated dose dispensing robots and the context in which they are used within the health system. My general understanding, coming from a country that does not have automated dose dispensing for primary care and based on the literature and conference presentations, is that the process of drug dispensing in the pharmacy is automated, with the meds required at any one time packaged and duly identified. The paragraph implies i) that the patients have a robotic dispensing device in their homes and ii) that healthcare providers would normally come to their homes to give them their daily medications. More information is needed on how the health system works, who pays for these dispensers and whether all of the literature reviewed relate to home use of ADD.”

Similarly, the abstract needs to better reflect the fact that dispensing is taking place at home. Again on reading it for the first time, I noted, “Abstract lines 3-4: “little is known about how ADD affects the patient’s daily life, getting a medication dispenser to do the job of a healthcare professional.” What is the rationale for investigating how medicines dispensed and packaged by an automated process rather than by health professionals will affect patients’ daily life? If they are in hospital, the packages of medicine will either be given to them to take at the correct time, or they will have been provided with a supply of packages of medicines as would happen when they are medicating at home. How does the health professionals lack of involvement in dispensing affect patients taking their meds at home?”

The second relates to the methods and results. According to the Methods, two databases were searched and 11 papers were identified for review. These are reported in Tables 1 and 2. But, in the section, Results, Characteristics of Included Studies, paragraph 5, “10 articles located as grey literature were included in the review as well.”  No description of the grey literature search is provided in the methods section. This implies that 21 papers were reviewed. If so, no results are provided for these ten papers in Tables 1 and 2, and figure 1 is incomplete.  

To further complicate matters, there are a total of 22 references in the paper, at least 5 of which 1, 8, 9, 10 and 11 do not appear to have anything to do with automated dispensing. Where then are the additional ten papers?

The Methods are not described in enough detail to be able to replicate the review.

General queries:

The title does not adequately reflect the content of the paper as automated dose dispensing of medicine can and does take place in pharmacies and hospitals.

Title: the word “off” should be “of”

Abstract: there is no stated aim of the study in the abstract.

Results: as there were only 11 papers reviewed the number of papers reported in the different sections of the results should be presented as a number and not just a percentage.

Introduction, para 1, lines 3-5: “50% of people with chronic diseases do not take their prescribed medication as recommended, and in developing countries the number can be lower due to paucity of health resources.” Based on the argument presented the percentage of people in the developing who fail to take their medications as recommended would be higher and not lower, as stated in the sentence?

Materials and Methods, para 2 and 3: inclusion and exclusion criteria. In paragraph 2, “…patient’s perception on automated dose dispensation in… the hospital” is an inclusion criterion, but in para 3, “automated dose dispensation in hospitals and pharmacies were excluded.” According to the abstract, “literature in a hospital setting were excluded”.

Methods, Study selection: what is the “literature directory application that provides the systematic overview of references to the scoping review” referred to in the text?

Figure 1: given that Embase contains almost all of the content of PubMed, it is surprising that only 35 duplicates were found.

Prisma Figure: presumably, this is the figure generated by the Covidence software, but it does not show how many papers were found by database searching. According to the data in appendix A, 105 papers were found in EMbase (search #27) and either 126 (search #25) or 132 (search #28) in PubMed. How then was the figure of 229 papers presented in the Prisma figure reached?

Appendix A: for the PubMed search, there are two searches numbered 19.

Appendix A: for PubMed, search 27 refers to combining searches #34 and # 37 which are not listed.

Table 2, Bardage et al., line 9: “three thirds of physicians and nurses…” Three thirds is one, or in the sense of the sentence, “all”.

References: the format of authors’ names for reference 10 is inconsistent with the rest of the references.

References: the authors’ names are not given for reference 14.

Author Response

To reviewer 1: Thank you very much for your important response. The detailed response is in the attachment.

Reviewer 2 Report

This review focuses on user experience and perception on Automated dose dispensing (ADD) systems for patients and Health care professional’s.

The work is fairly structured and the methods of investigation are clear despite the limited literature included in the study.

I strongly believe that the authors should integrate in this paper an analysis of the different types of existing ADD, dividing them by target (home or professional usage, age and cognitive wellbeing of patients, etc.), technology and further discuss the results.

Author Response

To reviewer 2: Thank you very much for your important response. The detailed response is in the attachment. 

Reviewer 3 Report

The manuscript “User perception off automated dose dispensed medicine: A scoping review” aims at reviewing available literature on users’ own perception of having an ADD robot. Although the general topic is relevant and up to date, the practical and theoretical implications for this research work remains low (as usually seen in scoping reviews with a very small amount of included papers).

General comments:

Please do not use keywords from the title.

The abstract should be restructured to include a clear rationale behind the study and a concise description of method, results, and conclusion.

It is redundant to write 1) AND background.

Due to inconsistencies and typos, the paper is partly hard to follow. E.g. the terminology is partly confusing when introducing an abbreviation, which is then used as Automated Dose Dispensing Robots (ADD), ADD, or ADD robots; patients experience vs patient experiences and similar ones.

Please explain the term prisma.

Please check expressions such as “Half of the six studies”!

It seems that in the table the abstract of the retrieved papers are c/p. Please rephrase and shorten.

I may have over-read this information, but I think the time period of the search is not mentioned.

Please check referencing style: e.g. [16-17][19].

The reference list should be checked and streamlined to avoid references like this: Covidence. Covidence. 2016. https://www.covidence.org/.

The final conclusion is very short and weak and could be substantiated accordingly.

Author Response

To reviewer 3: Thank you very much for your important response. The detailed response is in the attachment. 

Round 2

Reviewer 1 Report

Thank you for the opportunity to review the revised paper, which has been improved. Unfortunately, some of the revised sections raise further queries.

Figure 1: 14 studies were excluded. One of the reasons given is “4 double”. What does this mean? They couldn’t have been duplicates as the first step was to remove the duplicates.

Characteristics of the included studies: the percentages have been changed to numbers. Unfortunately, the numbers don’t add up correctly. Lines 140-142 document 10 papers, but the seven should be six, and thus only nine of the 11 papers are documented here.

Line 97: “line 113 has been deleted” should be deleted.

Conclusions: the second to last sentence is tautologous. Remove, “with only a limited number of publications so far”.

English editing is still required. Examples being, lines 54, 55 and 82.

Reviewer 3 Report

I thank the authors for substantially modifying their manuscript.

Author Response

Thank you for your acknowledgement.